# Ultrasound Defect Localization in Shell Structures with Lamb Waves Using Spare Sensor Array and Orthogonal Matching Pursuit Decomposition

**DOI:** 10.3390/s21238127

**Published:** 2021-12-04

**Authors:** Weilei Mu, Yuqing Gao, Guijie Liu

**Affiliations:** 1Engineering College, Ocean University of China, Qingdao 266100, China; muweilei@ouc.edu.cn (W.M.); gaoyuqing@stu.ouc.edu.cn (Y.G.); 2Suzhou Academy, Xi’an Jiaotong University, Suzhou 215123, China

**Keywords:** Lamb wave, modal separation, damage imaging

## Abstract

Lamb waves have multimodal and dispersion effects, which reduces their performance in damage localization with respect to resolution. To detect damage with fewest sensors and high resolution, a method, using only two piezoelectric transducers and based on orthogonal matching pursuit (OMP) decomposition, was proposed. First, an OMP-based decomposition and dispersion removal algorithm is introduced, which is capable of separating wave packets of different propagation paths and removing the dispersion part successively. Then, two simulation signals, with nonoverlapped and overlapped wave packets, are employed to verify the proposed method. Thereafter, with the proposed algorithm, the wave packets reflected from the defect and edge are all separated. Finally, a sparse sensor array with only two transducers succeeds in localizing the defect. The experimental results show that the OMP-based algorithm is beneficial for resolution improvement and transducer usage reduction.

## 1. Introduction

Ultrasonic guided waves have become the preferred tool in nondestructive testing and structural health monitoring (SHM). Lamb waved are one of the most commonly used ultrasonic guided waves in damage detection, due to their intrinsic advantages [1,2]. Lamb waves can propagate in a thin shell structure for a long distance with low energy loss. At the same time, they are sensitive to different types of damage such as cracks and corrosion [3]. Usually, a change of sensing signal amplitude or energy can be considered to be a change of wave-guided material, which is generally caused by damage.

In most mechanical structures, such as airplanes, ships, and deep submersibles, there are a lot of covering skins fixed by rivets. (e.g., Figure 1). Although the skins do not bear the structural gravity and workload, they play a great role in structure safety. There have been serious airline accidents caused by damage to skins. Usually, the damage is in the form of fatigue cracks in the rivet area. Therefore, the defects are mostly found at locations close to the edge. In this case, generated Lamb waves propagate to the defect and edge zone, and are reflected by the crack and edge. Consequently, the sensing signals contain the direct arrival wave, and the reflection waves from crack and edge. The wave signals may overlap each over, which makes signal explanation rather difficult. However, if these individual waves can be decomposed, it is beneficial to localize the damage, and moreover, it helps to reduce the number of deployed sensors. In this paper, we try to decompose the overlapped signals from the collected signal of two sensors. Ideally, it is best to deploy only one self-sensing sensor, which can generate and sense simultaneously.

Due to the dispersion nature of Lamb waves, waves of different frequencies will propagate with different speeds, which causes the wave packet to widen during propagation. Therefore, the dispersed waves of each path are overlap with each other more easily than without dispersion. Moreover, there are at least two modes of Lamb waves at any frequency, due to their multimodal nature, which makes signal processing more difficult [4]. Therefore, signal processing is essential for a Lamb wave-based SHM system. Wang et al. proposed a damage imaging algorithm based on time-of-arrival (TOA), in which they innovatively defined the possible location of the defect to be an ellipse with excitation and reception sensors as the focal points [5]. Following from that, Hall and Michaels improved the TOA algorithm using adaptive weighting coefficients, making the location imaging more accurate [6]. Another commonly used imaging method is based on time difference of arrival (TDOA) to different sensors; mathematically, the possible location of the defect seems to be a hyperbolic line. When using the TDOA algorithm, the number of sensors increases, but the imaging quality is not significantly enhanced [7]. Perelli et al. proposed a damage localization based on warped frequency transform (WFT) [8], which decomposes each wave packet from the spectrum of WFT and localizes the damage using these decomposed wave packets. However, the decomposed wave, not the original excitation wave, is dispersed, which may reduce the localization resolution. If damage occurs near the edge, there are multiple simultaneous propagation paths. Therefore, Ebrahimkhanlou et al. initially used a priori information of scattered wave fields to confirm each wave path. Then, the start and end points are taken as the focal points, and the TOA method is used to localize the damage. However, they used the original signal to confirm the possible localization line [9]. To improve the localization resolution, in this paper, it is similarly necessary to clarify the path of scattered waves. Moreover, accurately separating individual wave packets is also important.

In the history of scattered wave decomposition, decomposition methods based on dictionaries have gained great popularity. Alguri et al. constructed a dictionary containing wave signals of different travel distance using experiments, and then reconstructed the scattered waves from the online monitoring data [10]. Golato et al. built a dictionary consisting of multi-mode scattered Lamb waves and solved the sparse reconstruction problem of Lamb wave signals [11]. However, the above-mentioned methods above did not accomplish modal separation and elimination of the dispersion. Caibin Xu et al. built a dispersion signal dictionary using the dispersion curves to sparsely decompose the recorded dispersive guided waves, and innovatively obtained dispersion-compensated guided waves by using a non-dispersion signal dictionary [12]. Furthermore, they furthered an interesting method to obtain the dispersion curve of an unknown material, in which they acquired the unit pulse response of the measured structure, and interpolated the transfer function from the frequency domain into the wavenumber domain [13]. After this, they systematically proposed the construction method of over-completed dictionaries, which contained dispersive atoms and nondispersive atoms [14].

Recently, Caibin Xu et al. have proposed a focusing classical multiple-signal classification (MUSIC) algorithm based on the virtual time reversal technique, which can localize multi-defects without baseline subtraction. The proposed method is valuable for the practical use of Lamb wave-based structural health monitoring [4]. The matching pursuit (MP) algorithm, proposed by Mallat and Zhang, can decompose the signal into a combination of waveforms selected from the dictionary [15]. However, the vertical projection of the signal onto selected atoms is non-orthogonal, which makes the result of each iteration not optimal but suboptimal, requiring more iterations to achieve convergence. In this paper, we propose a guided-wave imaging method based on sparse reconstruction, which uses an analytical guided-wave propagation model to generate scattered signals for different propagation distances and modes, known as atoms of the dictionary. We constructed a dictionary containing wave signals of different travel distances by equations, and built a nondispersive dictionary by equations. In addition, then, the overcomplete dictionary is used to match the collected Lamb wave signals under the assumption of signal sparsity, which means there are only a few damage-related signals in the collected signal. Indeed, the damage occurs at only a few discrete locations in the structure. The orthogonal matching pursuit (OMP) algorithm is used to separate each wave component. The propagation distance is related to the labels of the selected atoms. The OMP algorithm orthogonalizes all the atoms needed at each step of the decomposition, and converges faster [16].

The rest of the article is organized as follows. Section 2 describes in detail how to construct the dictionary using the Lamb wave scattering model and the procedures of the OMP algorithm, and how to remove dispersion simultaneously. In Section 3, numerical simulations are carried out, nonoverlapped and overlapped signals are decomposed, and dispersion is removed by the OMP algorithm to verify its correctness. In Section 4, experimental studies are implemented to validate the proposed method. Finally, conclusions are drawn in Section 5.

## 2. Methodology

### 2.1. Introduction of OMP

If the signal is sparse on some basis functions during decomposition, it means that the energy of the signal gathers on a few basis functions. Therefore, the signal can be represented by these basis functions and their coefficients [17].

Suppose that given a dictionary $D=\{a_{1},a_{2},a_{3}\ldots \ldots a_{L}\}$, L≫N. For arbitrary signals *y*∈*R^N^*, select *K* atoms in the dictionary *D* and make a *K*-term approximation to *y*.
(1)$$y_{K}=\sum_{i\in I_{K},\left|I_{K}\right|=K}\langle y,a_{i}\rangle a_{i}$$
where *I_K_* is the subscripts set of the *K* atoms, corresponding to the first *K* maximum coefficient of $\left|{y,a}_{i}\right|$.

In other words, it prefers selecting the set of atoms with the sparsest decomposition coefficients from various possible combinations of dictionary atoms, which also reflects the idea of sparse signal decomposition.

### 2.2. Overcomplete Dictionary Construction

In an infinitely large thin plate, the structure response at a distance of *d* from the excitation can be expressed as [18]
(2)$$u_{n}^{A\mathrm{or}S}(t)=\sqrt{\frac{1}{d}}\frac{1}{2\pi }\int_{-\infty }^{+\infty }S(\omega )e^{j\omega t}e^{-jk_{n}^{A\;\mathrm{or}\;S}(\omega )d}d\omega$$
where *u(t)* is the time domain response signal of *n-th* symmetric mode (S) or anti-symmetric mode (A), *t* is the travel time, *ω* is the angular frequency, *S(ω)* is the frequency domain representation of the excitation signal *f(t)*, *k(ω)* is the wave number of the corresponding Lamb wave mode, and *j* is the imaginary unit. The frequency representation of Equation (2) can be written as
(3)$$U(\omega )=\sqrt{\frac{1}{d}}S(\omega )e^{-jk(\omega )d}$$

The same as above, if there is a through-thickness damage in the plate, then the propagation distance from the excitation source to the damage then to the receiver is *ds*. Therefore, the received scattered signal *Y(ω)* in the frequency domain can be expressed as
(4)$$Y(\omega )=\alpha (\omega )\beta (\omega )\sqrt{\frac{1}{d_{s}}}S(\omega )e^{-jk(\omega )d_{s}}$$

When there are multiple damages in the plate, the corresponding scattered signal can be approximated as a linear superposition of each individual case, without considering the scatter phenomenon between any two damages [18,19]
(5)$$Y(\omega )=\sum_{i}\alpha_{i}(\omega )\beta_{i}(\omega )\sqrt{\frac{1}{d_{s}{}^{i}}}S(\omega )e^{-jk(\omega )d_{s}{}^{i}}$$

In the above equation, *α_i_(ω)* is the scattered coefficient from the excitation source to the damage of the *i-th* damage; *β_i_(ω)* is the scattered coefficient from the damage to the receiver of the *i-th* damage, and *d_s_^i^* is the propagation distance of the *i-th* propagation path. From Equation (2), it can be seen that the signal is obtained by multiplying the atoms in the dictionary with their coefficients, so the value of the scattered coefficient *α_i_(ω)* will be reflected in the coefficient matrix *θ*. From this, the *i-th* column of the dictionary *D* of A or S mode can be expressed as
(6)$$a_{i}=F^{-1}\left\{\sqrt{\frac{1}{d^{i}}}S(\omega )e^{-jk_{n}^{AorS}(\omega )d^{i}}\right\}$$
where *d^i^* is the propagation distance of the *i-th* atom, *F^−1^{·}* represents the inverse Fourier transform. Thus, the single-mode dictionary can be obtained in the form [12]
$$D^{A\;\mathrm{or}\;S}=\left[\begin{array}{c} a_{1}{}^{1}a_{2}{}^{1}\cdots a_{i}{}^{1}\cdots a_{L}{}^{1}\\ a_{1}{}^{2}a_{2}{}^{2}\cdots a_{i}{}^{2}\cdots a_{L}{}^{2}\\ \vdots \;\vdots \;\ddots \;\vdots \;\ddots \vdots \\ a_{1}{}^{N}\;a_{2}{}^{N}\ldots a_{i}{}^{N}\cdots a_{L}{}^{N} \end{array}\right]$$
where the aim is the value of the m-th sampling point in the i-th atom.

### 2.3. Nondispersive Dictionary

The expressions of wave number, phase velocity, and group velocity are respectively [14]
(7)$$\begin{array}{l} k_{a}(\omega )=\frac{1}{C_{p}(2\pi f)}\omega \\ C_{p}(\omega )=\frac{\omega }{k_{a}}=C_{p}(2\pi f)\\ C_{g}(\omega )=\frac{d\omega }{dk_{a}}=C_{g}(2\pi f) \end{array}$$

It is proved that if the wave number of the Lamb wave is linearly related to the frequency, the envelope of the response signal will be the same as the envelope of the excitation. In other words, if the wave number of Lamb wave is linear with respect to frequency, it will not be dispersive. Therefore, the propagation distance can be calculated by the group velocity of central frequency.
(8)$$x_{i}=C_{g}(2\pi f_{c})t_{i}$$

Bringing the above equation into Equation (3) and neglecting the amplitude change due to signal propagation, we can obtain the nondispersive signal *y_i_(t)*.
(9)$$y_{i}(t)=\frac{1}{2\pi }\int_{-\infty }^{+\infty }S(\omega )e^{j(\omega t-k_{a}x_{i})}d\omega =\frac{1}{2\pi }\int_{-\infty }^{+\infty }S(\omega )e^{j\omega (t-t_{i})}d\omega =f(t-t_{i})$$

Equation (9) shows that the nondispersive signal is equal to the excitation signal with a phase shift, so the nondispersive dictionary *D_n_* can be built.
$$D_{n}^{A\;\mathrm{or}\;S}=\left[\begin{array}{c} f_{1}(t-t_{1})\;f_{1}(t-t_{2})\cdots f_{1}(t-t_{i})\cdots f_{1}(t-t_{L})\\ f_{2}(t-t_{1})\;f_{2}(t-t_{2})\cdots f_{2}(t-t_{i})\cdots f_{2}(t-t_{L})\\ \vdots \;\vdots \;\ddots \;\vdots \;\ddots \;\vdots \\ f_{m}(t-t_{1})\;f_{m}(t-t_{2})\cdots f_{m}(t-t_{i})\cdots f_{m}(t-t_{L})\\ \vdots \;\vdots \;\ddots \;\vdots \;\ddots \;\vdots \\ f_{N}(t-t_{1})\;f_{N}(t-t_{2})\cdots f_{N}(t-t_{i})\cdots f_{N}(t-t_{L}) \end{array}\right]$$
where *f_m_(t-t_i_)* is the value of the *m-th* sampling point of the nondispersive signal with time shift *t_i_*.

### 2.4. OMP-Based Decomposition and Dispersion Removal Algorithm

Assuming that *y_s_* contains *r* wave packets, the received signal *y_s_* can be represented by a linear combination of its atoms [18]
(10)$$y_{s}=D\theta +n$$
where *D*∈*R^N×M^*
*(M ≥ r)* is the overcomplete dictionary, *θ*∈*R^M^* is the coefficient column vector, and *n*∈*R^N^* is the residual noise term.

The above equation is an underdetermined equation. If the number of atoms *M* is large enough and the travel distance *d_s_^i^*
*(1 ≤ i ≤ r, i*∈*N^+^)* of the scattered signal is completely covered by the distance *d^i^*
*(1 ≤ i≤M, i*∈*N^+^)* corresponding to the given atoms, then, the scattered signal can be sparsely decomposed with the dictionary *D* by the OMP algorithm.

OMP is an iterative algorithm capable of selecting the best fitness atoms from the overcomplete dictionary for signal reconstruction. The OMP algorithm is described as [20]
(1)Initialization process. Determine sparsity degree *K* which means the number of potential wave packets. Build an overcomplete dictionary *D* as:
$$D=[a_{1},a_{2},a_{3}\ldots a_{i}\ldots a_{L}]$$
where *ai* indicates the *i-th* atom in the dictionary.

(2)Orthogonal matching. Find the column *a_λ_* in the dictionary according to the product value *θ_λ_* of acquisition signal *y* and *a_λ_*. Then, record the product value *θ_λ_*, known as matching coefficients. $a_{\lambda }={\mathrm{argmax}}_{i=1,2\ldots N}\left|\langle y,a_{i}\rangle \right|$, *λ* indicates that the atom is the *λ-th* column in the dictionary, and $\theta_{\lambda }=\max_{i=1,2\ldots N}\left|\langle y,a_{i}\rangle \right|$.(3)Update and iteration. Update the solution set $\theta =\theta \cup \{\theta_{\lambda }\}$, and update the residual signal by subtracting the selected atoms from the signal of last iteration.


$$y=\min{\left\|y-a_{\lambda }\theta_{\lambda }\right\|}_{2}^{2}$$


(4)Termination judgment. Determine whether the number of iterations is greater than *K*. If it is not satisfied, execute the matching and update procedures again.

It should be noted that the solved *θ* is a column vector with mostly zero elements, when using the overcomplete dictionary, and the non-zero element *θ_λ_* of *θ* represents the scattering coefficient of the *λ-th* atom in the dictionary with travel distance *L**_λ_*. Each scattering wave packet with a unique travel distance in the collected signal can be recovered using the above equation and the overcomplete dictionary *D*. The propagation distance of the scattering wave packet can be visually expressed by the travel distance of the *λ-th* atom. With the same column label, the collected signal can be represented as the combination of nondispersive wave packets using the nondispersive dictionary. Sparsity degree *K* determines the number of atoms selected from the dictionary. When the value of *K* is small, some wave packets may not be matched. When the value of *K* is large, matching performance will be better but the amount of calculation is greatly increased. Therefore, the total number of wave packets *m* in the signal needs to be predicted, and in general, the value range of *K* is *2m ≥ K ≥ m*. The procedure of the proposed method is shown in Figure 2.

## 3. Methodology Verification

### Numerical Simulation

To verify the performance of the OMP-based dispersion removal approach, numerical simulations are first provided, including a nonoverlapped case and an overlapped case.

Abuqus CAE was adopted to simulate Lamb wave propagation, and a simulation model with the dimensions 1250 mm × 1250 mm × 2 mm (length × width × thickness) was conducted. At the edge regions, the wave absorption layer was built to eliminate the reflected wave. Moreover, two different actuators with the locations *A* (100,400) and *B* (300,400), and one receiver with location *C* (800,400) were set. Therefore, two different wave-guided paths were simulated in this case, and the distances between the actuator and receiver were different. Material parameters of the simulation model are listed in Table 1.

In the numerical simulation, the larger the grid size, the less computational consumption. However, with the increase of the grid size, simulation error decreases. On the contrary, a smaller size might increase the computational consumption exponentially. Therefore, element size is usually limited to one-tenth of the wavelength. In this case, the central frequency of the excitation signal was 100 kHz, and the grid size in this study is set as 1 mm. According to Nyquist’s theory, the acquisition frequency must be at least twice the maximum frequency component. In the simulation and experiment, the acquisition frequency was 2.5 MHz for full frequency range acquisition [21].

A five-cycle sinusoidal signal modulated with Hanning window was selected as the excitation signal, it can be expressed as follows:(11)$$y=A\sin(2\pi f_{c}t)\times (1-\cos(2\pi f_{c}t/N))$$
where *A* is the amplitude of the sinusoidal signal, *N* denotes the cycle number of the modulated signal, *f_c_* represents the central frequency, and *t* is time serial ranging from zero to *N/f_c_*.

Since the center frequency used is below the cutoff frequency of S1 and A1 modes, only two fundamental Lamb wave modes A0 and S0 are considered [22]. As shown in Figure 3a, the dispersive signal contains a total of four wave packets, which are two mode components for two propagation paths. By OMP algorithm, the original signal can be separated to four wave packets. The two separated S0 and A0 mode wave packets are as shown in Figure 3b,c respectively.

The A0 and S0 modes can be easily separated by the previously built single-mode dictionary, and it can be seen that the A0 mode has undergone more serious dispersion, as shown in Figure 3c. Reconstruction of the separated modal signals using a nondispersive dictionary can better eliminate the dispersion, as in Figure 4.

In the modal separation of Lamb waves, the sparse decomposition of the dispersive signal is based on the corresponding modal dictionary, which must contain all the corresponding atoms to fully compensate all the dispersive modes. Otherwise, the dispersive signal will be incompletely compensated [23]. In signal reconstruction using only the S0 modal dispersion dictionary, the dispersive signal is recovered for the S0 mode component, while the A0 mode component is not recovered, as shown in Figure 5. This is because there is no atom corresponding to the wave packet of A0 mode in the S0 mode dictionary. In conclusion, the proposed approach based on sparse decomposition cannot handle the case with an incomplete dictionary. In other words, the maximum propagation distance of the atom signal in the single-mode dictionary must be larger than that of the signal to be processed, and the same requirement is suitable for the atom signal in the nondispersive dictionary. In practice, the time information depending on the length of the dispersive signal acquisition can be known in advance, and then the dispersive and nondispersive dictionaries can be built to cover all possible wave packets.

Usually, there are many wave packets from different propagation paths collected by the acquisition system simultaneously. Apparently, there are wave packets that overlap each other. For further verification of the proposed algorithm, the exciter positions are set to *A* (250,400) and *B* (300,400), and the receiver position remains unchanged. In contrast to Figure 5, there seems to be only two wave packets of two mode components, as shown in Figure 6, due to the overlap phenomenon.

However, when sparse decomposition of the signal is performed, it still requires two atoms from the A0 dictionary and two atoms from the S0 dictionary to match it. In the same way as above, the matched atoms are used to represent the wave packets of the signal, and finally the dispersion is removed using the nondispersive dictionary. The procedure is shown in Figure 7. It turns out that this approach is still able to separate the wave packets, even if they are overlapped.

## 4. Experimental Verification

### 4.1. Sparse Sensor Array-Based Localization Method

The PZT wafer A and B with dimensions 13 mm × 0.5 mm (diameter × thickness) are mounted at (*x_A_, y_A_*) and (*x_B_, y_B_*), respectively. Therefore, the propagation distance of the primary scattered wave (as shown in Figure 8) that is reflected only by the defect can be described as
(12)$$d=\sqrt{{\left(x-x_{A}\right)}^{2}+{\left(y-y_{A}\right)}^{2}}+\sqrt{{\left(x-x_{B}\right)}^{2}+{\left(y-y_{B}\right)}^{2}}$$

Although the defect location is unknown in practical detection, a through-thickness hole is fabricated to make an analogy to the defect and drilled at (*x_c_,y_c_*) for this experiment. In the process of defect localization, at least three different propagation distances need to be determined for a more accurate localization [24]. Traditionally, each propagation distance is confirmed by a pair of PZT wafers. Hence, at least three PZT wafers are needed. In addition to the direct arrival wave and the primary scattered waves, there are also secondary scattered waves reflected from the edges and defects. Figure 9a indicates the signal via path *a-b*, excited by the exciter A and reflected by the defect. Moreover, sensor B will also receive signal from paths *c-d, e-f*, and *e-d*, with two reflections by the defect and the edge. The latter two scattered waves are relatively weak compared with the first one, and are not considered here.

The incident angle into the edge is equal to the reflection angle that is *θ_i_ = θ_r_*, so the scattered signal path *c-d* can be easily deduced out by the mirror point of sensor B, and its propagation distance is
(13)$$D_{cd}=d_{c}+{d^{\prime }}_{d}=\sqrt{{\left(x-x_{A}\right)}^{2}+{\left(y-y_{A}\right)}^{2}}+\sqrt{{\left(x-{x^{\prime }}_{B}\right)}^{2}+{\left(y-{y^{\prime }}_{B}\right)}^{2}}$$
where *d_c_* denotes the distance of path *c* that is the distance from the excitation sensor A(*x_A_,y_A_*) to the defect; ${d^{\prime }}_{d}$ denotes the distance from the defect to the mirror point $B^{\prime }\left({x^{\prime }}_{B},{y^{\prime }}_{B}\right)$

With sensor B for excitation and sensor A for reception, as shown in Figure 9b, the propagation distance of the primary scattered wave is the same *D_ab_*. The propagation distance of the secondary scattered wave is
(14)$$D_{ij}=d_{i}+{d^{\prime }}_{j}=\sqrt{{\left(x-x_{B}\right)}^{2}+{\left(y-y_{B}\right)}^{2}}+\sqrt{{\left(x-{x^{\prime }}_{A}\right)}^{2}+{\left(y-{y^{\prime }}_{A}\right)}^{2}}$$
where *d_i_* denotes the distance of path *i*; ${d^{\prime }}_{j}$ denotes the distance from the defect to the mirror point $A^{\prime }\left({x^{\prime }}_{A},{y^{\prime }}_{A}\right)$ of sensor A(*x_A_,y_A_*).

According to analyses above, three wave paths are obtained with only two PZT wafers. The propagation distance determined by each pair of excitation–reception or excitation-to-mirror sensors can determine an ellipse with the sensor as the focus, as shown in Figure 10. The red ellipse in Figure 10a confirmed by PZT A to PZT B is the same as that in Figure 10b, due to the same propagation distance of A–B and B–A. Additionally, the blue ellipse and green ellipse confirmed by the primary scattered waves are combined with the red ellipse in Figure 10c to access the damage location.

When three ellipses are identified, their envelopes, or intersections, are the defect locations. As long as the measured wave packet has enough energy to be detected, the damage can theoretically be located with only one excitation [25]. However, in practice, the signal resolution and intensity are not high enough, so it is necessary to change the excitation to improve the imaging effect. PZT A and B will be taken as the actuator and sensor successively. When the plate is intact, the recorded signals are considered to be the reference signals. Ignoring the changes of wave guide material due to environment and damage, the wave signal varies only with respect to the scatter phenomenon of the damage [26]. The sparsity degree *K* can be set to be 8 at least, due to 4 propagation paths for S0 and A0 mode.

### 4.2. Experimental Setup

A square plate with dimensions 1250 mm × 1250 mm × 2 mm (length × width × thickness) was adopted as the experimental sample. Two PZT wafers A and B were mounted at (400, 150) and (700, 150), respectively. A prefabricated through-hole was located at (500, 50). The experimental setup consisted of a Tektronix AFG1022 arbitrary waveform generator, a DS2−8B data acquisition instrument and a smart AE charge amplifier. To coincide with the simulation case, the experimental parameters, such as the acquisition frequency and the excitation signal, were all the same as the simulation model.

As PZT A and B will be taken as the actuator and sensor successively, the actuator will be connected with Tektronix AFG1022, and the sensor will be connected with the charge amplifier which is connected to the data-acquisition device. At the same time, the waveform generator will be connected to the data acquisition device with an isolator to record the excitation and sensing wave packets simultaneously. The diagram of the experimental setup is shown in Figure 11.

Therefore, the reference-based monitoring method can be carried out as follows.

(1)under the intact condition, excite a five-peak sinusoidal wave modulated by Hanning window with a center frequency of 100 kHz from PZT A, collect the signals from PZT B. Then, excite modulated wave from PZT B, and collect the signals from PZT B. These signals are considered to be the reference signals.(2)in the practical monitoring, repeat the acquisition step above.(3)subtract the intact signal from the acquisition signal with defects, known as the state-relative signals.(4)decompose the state-relative signals into wave packets of a single mode by the OMP-based decomposition and dispersion compensation method.(5)localize the defect by sparse sensor array-based localization method.

### 4.3. Experimental Results

Under the intact situation, PZT A and B are taken as actuator and sensor, respectively. The collected signal was taken from PZT B for the sample, as shown in Figure 12a, which contains disturbed noise using the excitation signal, direct wave of S0 and A0 mode, and reflected wave using the edge. If the sparsity degree *K* is 8, the wave packets with relatively high amplitude are extracted. The recovery signal is shown in Figure 12a. Theoretically, there is also a reflected wave of S0 mode, although it is not determined visually. With the largest two matching coefficients *θ_λ_*, only the A0 mode waves are extracted, as shown in (b), because the amplitude of S0 mode wave is relatively low. Therefore, the OMP algorithm can remove the system noise well, because the predefined atomic functions are independent of the noise. Moreover, using the columns of the chosen atoms, it is easy to get the travel distance of the corresponding wave packets, which verifies the correctness of the method. Figure 12c shows the recomposed signal using the nondispersive dictionary to remove the dispersion.

After fabricating a through-thickness hole in the plate, we collect the signals from PZT A and B successively. Then, the received signals are subtracted from the corresponding signal when there is no defect, the residual signal is processed with the same process procedures, and the results are shown in Figure 13.

We decompose with the overcomplete dictionary and extract with the largest two matching coefficients. Only two scattered wave packets from two paths are obtained, as shown in Figure 13b. Therefore, wave packets on multiple paths are precisely separated, and the effect of measurement noise is significantly reduced. Then, we remove the wave dispersion by the non-dispersion dictionary, as shown in Figure 13c, which makes the energy more concentrated for damage imaging. To determine the case of the third propagation path, the signal is excited by PZT B and received by PZT A. The signal is processed using the same process and the results, as shown in Figure 13f–j. Meaningfully, Figure 13h indicates that the proposed decomposition approach can separate the overlapped wave packets, due to similar travel distance or the occurrence of dispersion. To improve the resolution of defect localization, the enveloped signals of the reconstructed waveform, as shown in Figure 13(d,e,i,j), are taken to generate the virtual wavefield for each exciter–sensor pair. Each wavefield characterizes the possible trajectory of the defect as an ellipse, and the energies of multiple wavefields are superimposed at the actual defect location [26,27], as shown in Figure 14.

Figure 14a shows the actual location of the PZT wafer A and B, and the mirror wafer of PZT A and B. With the state-relative signal, the imaging result, as shown in Figure 14b, presents a large energy concentration area, which is as about 15 times the size of the actual defect area. With the reconstructed signals, a relatively clear image can be obtained by filtering out unwanted noise signals, as shown in Figure 14c. With the final non-dispersion signal, a much higher-resolution image, as shown in Figure 14d, succeeds in localizing the defect with a similar energy-concentrated area. Therefore, with the OMP-based sparse reconstruction, the signals from the sparse sensor array can be separated.

## 5. Conclusions

To detect damage with as few sensors and as high resolution as possible, a method, using only two piezoelectric transducers and based on orthogonal matching pursuit (OMP) decomposition, was proposed. With the proposed algorithm, the wave packets reflected from the defect and edge are all separated. The experimental results show that a sparse sensor array with only two transducers succeeds in localizing the defect, and the OMP-based algorithm is beneficial for resolution improvement and transducer usage reduction. Some conclusions can be drawn as follows.

with the over-completed dictionaries of A0 and S0 mode, the OMP-based algorithm can separate wave packets from collected signals, even if the wave packets are overlapped. Thereafter, with the nondispersive dictionaries, the dispersion part is removed, which transforms the deformed wave packets to the original excitation signal.the wave packets reflected by the defect and edge are innovatively used for defect localization, which is the equivalent of mounting a virtual sensor at the mirroring position. The use of these multipath wave packets is beneficial for reducing the use of transducers.the dispersion-removed wave packets of multipath can localize the defect position and improve the resolution of defect localization.

A little further discussion is required. The method in this article is mainly used to locate a defect. When there are two defects, the energy is concentrated at two or more potential points. However, when there are too many defects, the signals overlap and separation is difficult, and this method is no longer suitable. 

The defect is located by first-order mode signals in the experiment. When other mode signals (e.g., A1, S1) are excited, it is possibly beneficial for high-resolution localization, if there are sufficient atoms of the corresponding mode in the over-completed dictionary. Although the method can separate overlapping signals, in practice, the OMP algorithm is no longer effective when the envelope peaks of the two signals completely overlap. When the travel distance of the wave packet is very short, the waveforms of A0 mode and S0 mode are similar to the excitation waveform, which may also produce matching errors. Therefore, the travel distance of the wave packet is suggested to be longer than 150 mm, when the A0 mode signal has been significantly scattered.

## Figures and Tables

**Figure 1 sensors-21-08127-f001:**
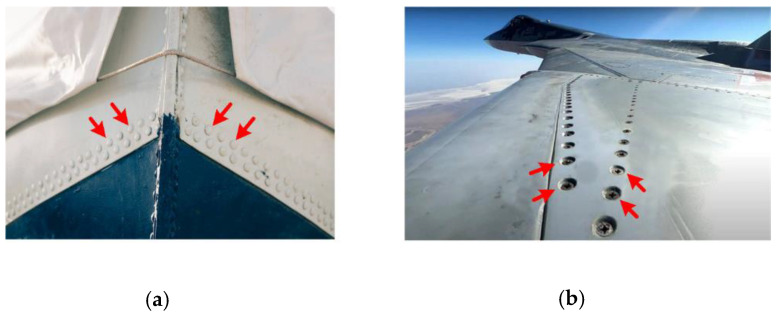
Skins of structures. (**a**) hull of ships and (**b**) aircraft skins.

**Figure 2 sensors-21-08127-f002:**
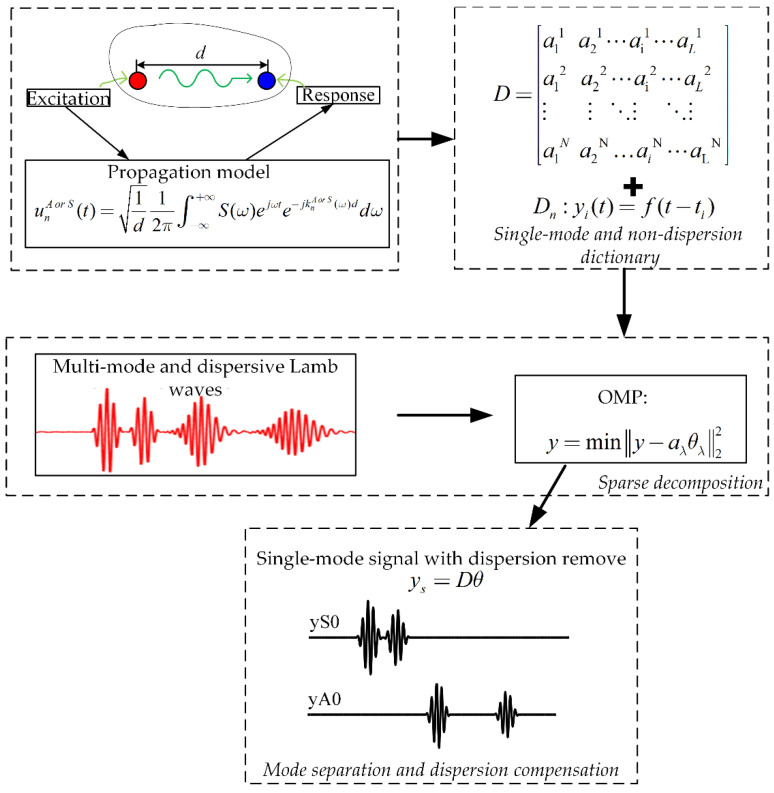
Schematic of the procedure of the proposed method.

**Figure 3 sensors-21-08127-f003:**
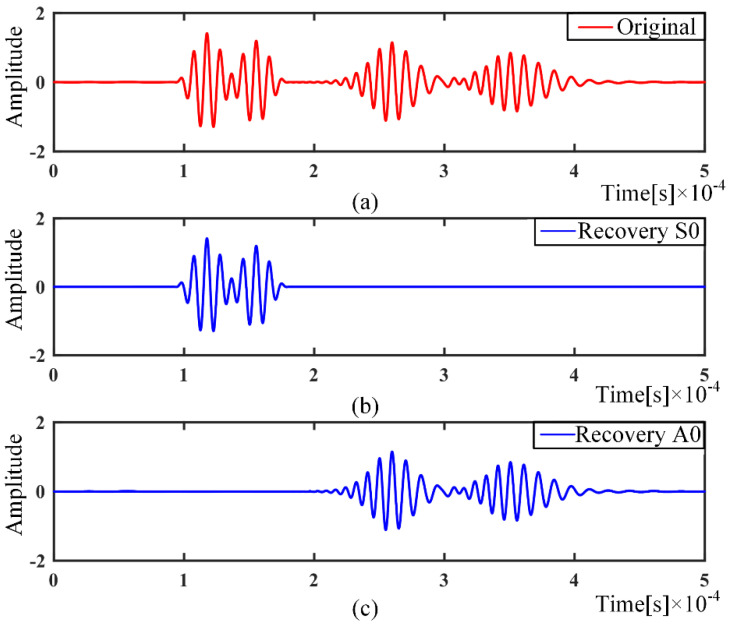
Mode separation: (**a**) Original signal; (**b**) Separated S0 mode dispersion signal; (**c**) Separated A0 mode dispersion signal.

**Figure 4 sensors-21-08127-f004:**
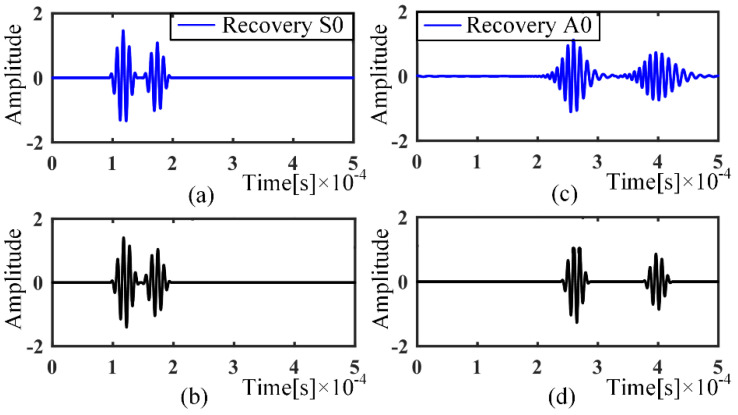
Dispersion compensation: (**a**) S0 mode signal; (**b**) S0 dispersion removal signal; (**c**) A0 mode signal; (**d**) A0 dispersion removal signal.

**Figure 5 sensors-21-08127-f005:**
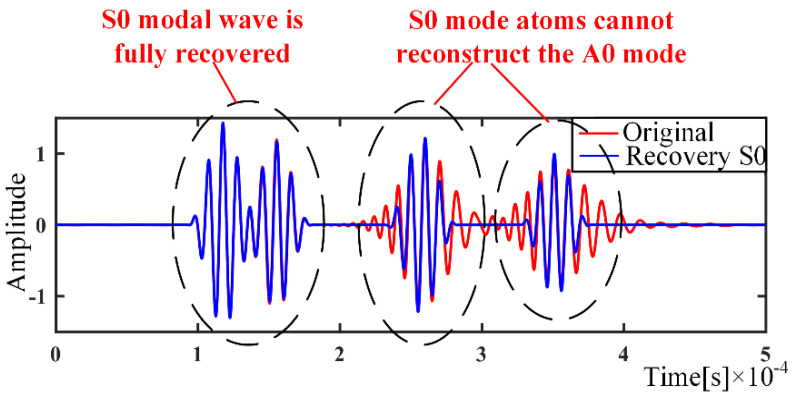
S0 modal recovery.

**Figure 6 sensors-21-08127-f006:**
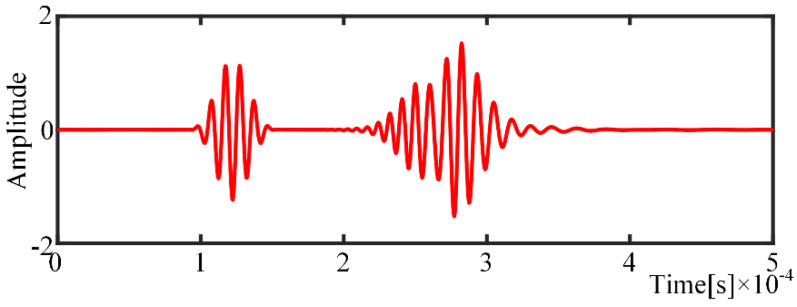
Overlapping wave packets.

**Figure 7 sensors-21-08127-f007:**
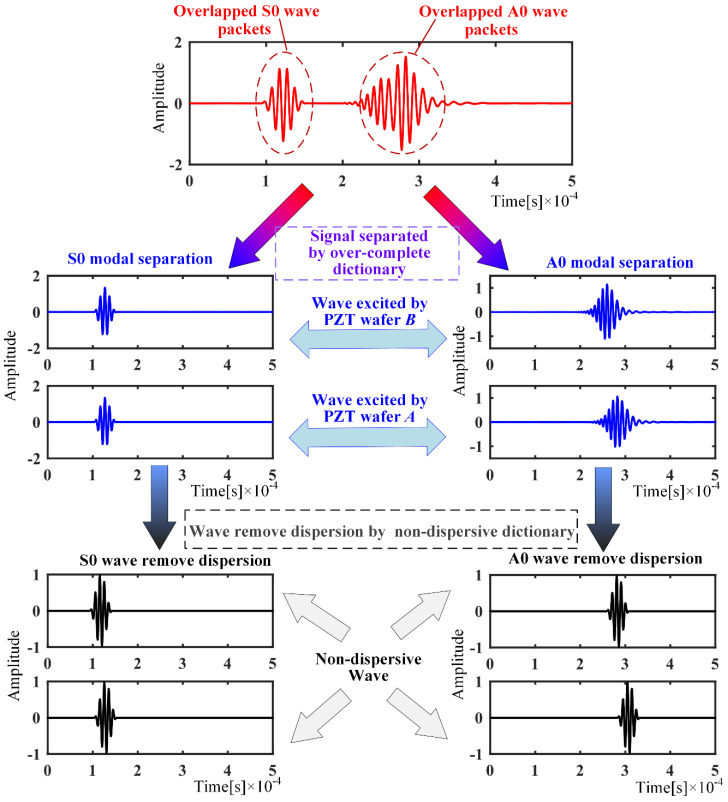
Separation of overlapped wave packets.

**Figure 8 sensors-21-08127-f008:**
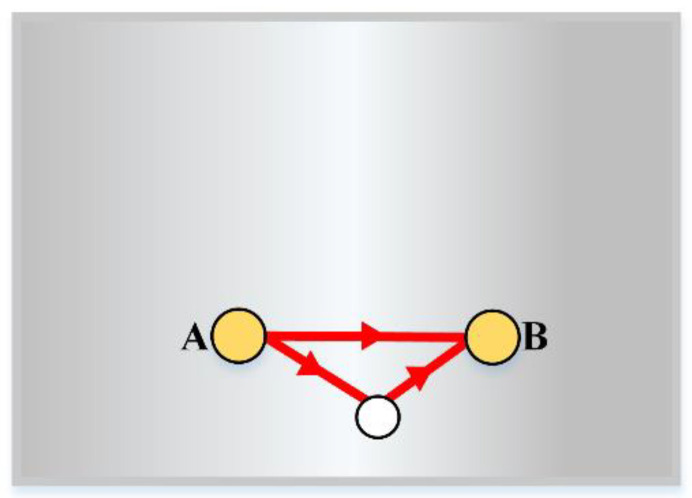
Experiment sample.

**Figure 9 sensors-21-08127-f009:**
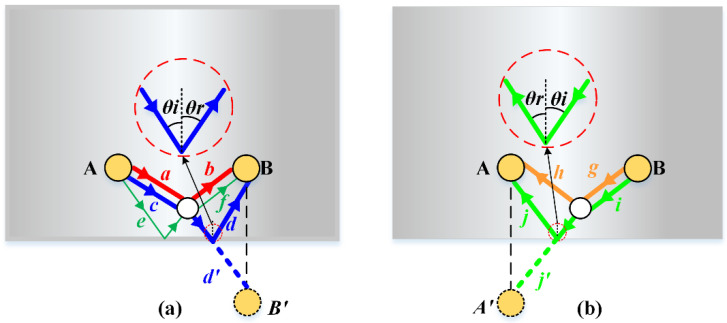
Guided-wave scattering path: (**a**) PZT A to PZT B; (**b**) PZT B to PZT A.

**Figure 10 sensors-21-08127-f010:**
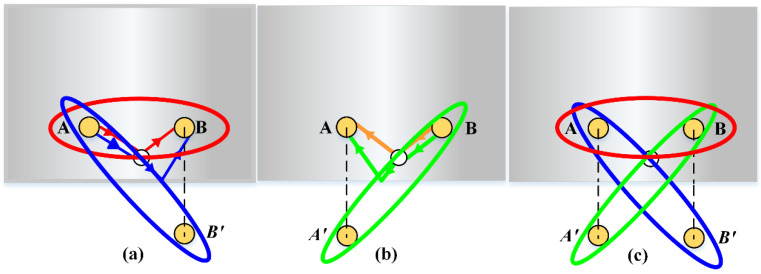
Elliptical trajectories: (**a**) PZT A to PZT B; (**b**) PZT B to PZT A; (**c**) combination.

**Figure 11 sensors-21-08127-f011:**
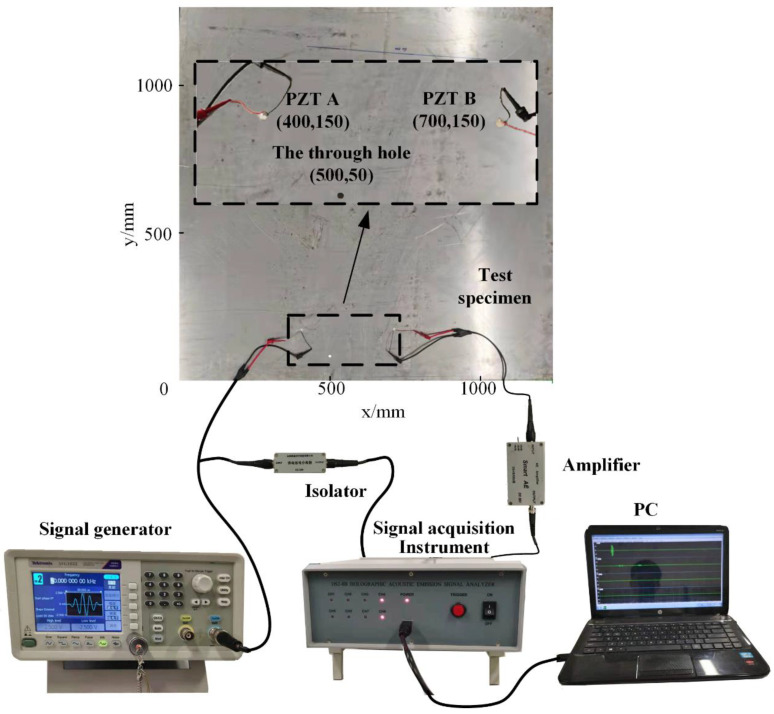
Experimental setup.

**Figure 12 sensors-21-08127-f012:**
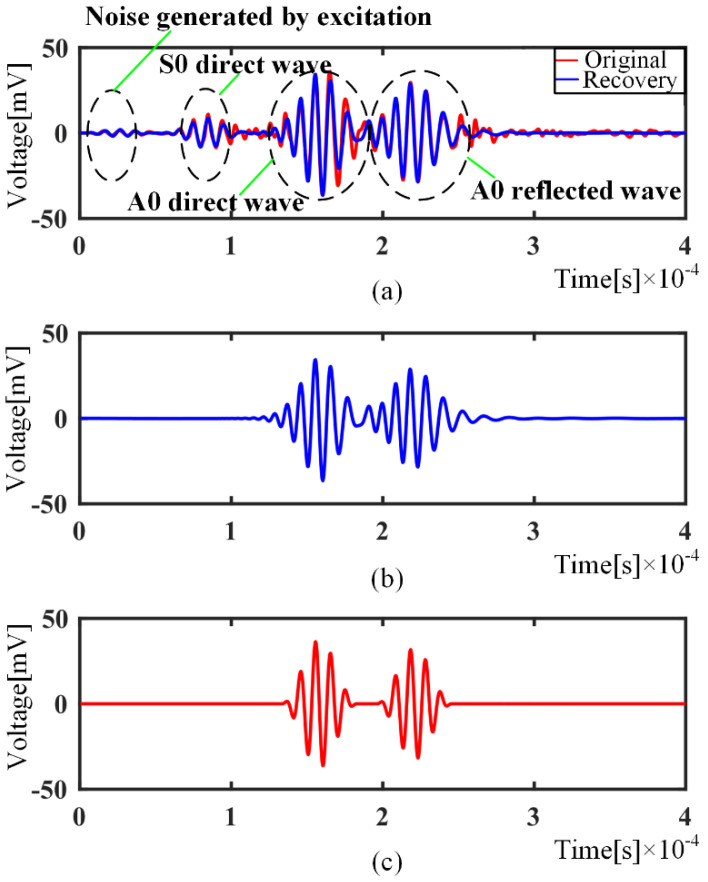
Defect-free signal processing: (**a**) Defect-free signal and recovery signal; (**b**) Recovery A0 with largest matching coefficients; (**c**) The result after dispersion removal.

**Figure 13 sensors-21-08127-f013:**
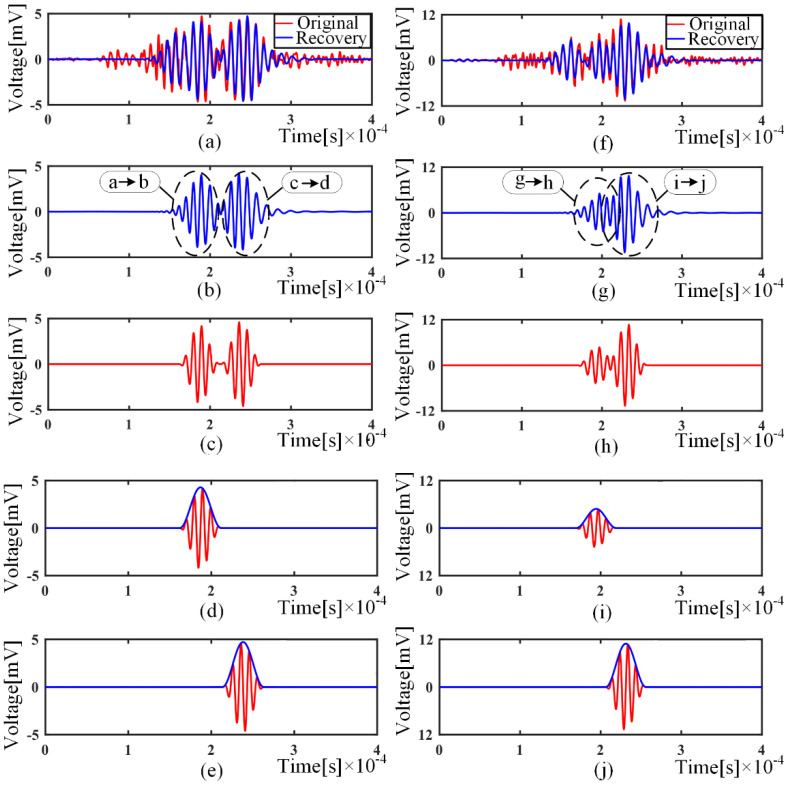
Residual signal processing: (**a**) Original residual signal and reconstructed signal; (**b**) Reconstructed signal with largest matching coefficients; (**c**) Reconstructed signal with dispersion removed; (**d**) Separated wave packet and envelope of path *a–b*; (**e**) Separated wave packet and envelope of path *c–d*; (**f**) Original residual signal and reconstructed signal; (**g**) Reconstructed signal with largest matching coefficients; (**h**)Reconstructed signal with dispersion removed; (**i**) Separated wave packet and envelope of *g–h*; (**j**) Separated wave packet and envelope of *i–j*;.

**Figure 14 sensors-21-08127-f014:**
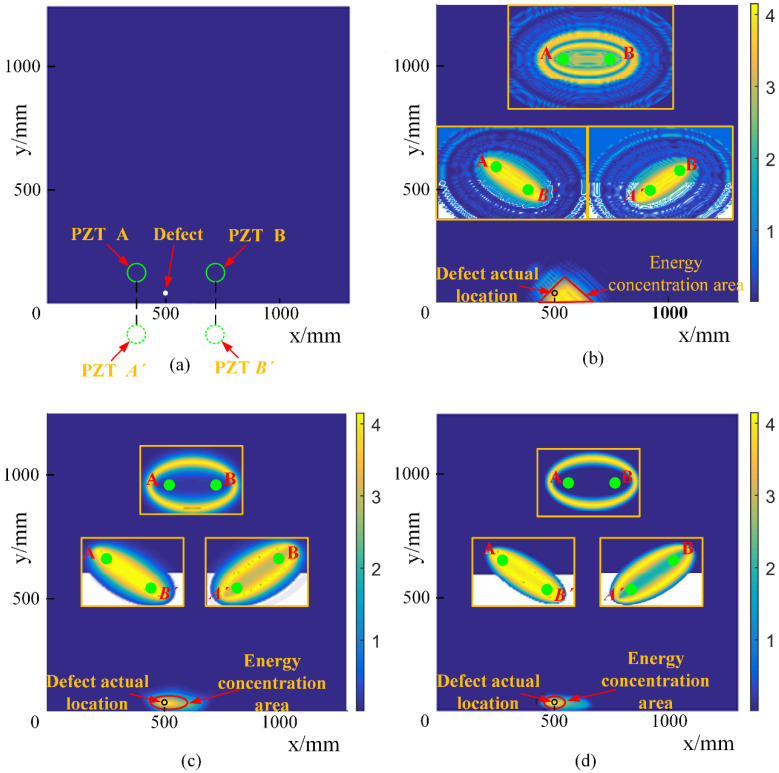
Damage imaging: (**a**) The location schematic of PZT; (**b**) The imaging result with original signal; (**c**) The imaging result with dispersion signal; (**d**) The imaging result with non-dispersion signal.

**Table 1 sensors-21-08127-t001:** Material parameters.

Material	Density (kg/m^3^)	Elastic Modulus (Pa)	Poisson’s Ratio
Q235	7800	2.1 × 10^11^	0.33

## Data Availability

The supplementary data and simulation programs involved in this paper will be uploaded by the first author on the website ofhttps://www.researchgate.net/profile/Weilei_Mu.

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
