# Peer review of "Ultrasound Defect Localization in Shell Structures with Lamb Waves Using Spare Sensor Array and Orthogonal Matching Pursuit Decomposition"

_sensors, 2021, doi:10.3390/s21238127_

Round 1

Reviewer 1 Report

This manuscript develops a defect localization method based on only two sensors. Both numerical simulation and experiments are implemented to test the proposed method. The article is within the scope of this journal. Overall, the paper can be published after major revision.

  1. It is hard to locate damage based on only two sensors on Lamb wave NDT and SHM. The authors claim that the proposed method can achieve defect localization using only two piezoelectric transducers. And the experimental results are all for a single defect. Does the method can locate two defects on a plate? It should be clearly point out the application scope of the method.

  1. In the introduction, the authors reviewed the methods for damage localization as well as dispersion effect. However, the authors here omitted a kind of Lamb wave damage localization methods called “MUSIC”. The authors are encouraged to provide the recent advances on damage localization methods based on MUSIC algorithms. For example, the following reference is related to the topic of MUSIC algorithms as well as dispersion effect:

-“A focusing MUSIC algorithm for baseline-free Lamb wave damage localization”, Mechanical Systems and Signal Processing, 164 (2022) 108242.

  1. In the introduction, the authors reviewed the scattered wave decomposition methods for Lamb waves. However, the methods developed in references [17, 18] and the one developed in the article titled “Sparse estimation of propagation distances in Lamb wave inspection” are all not mentioned. The three articles are also based on an analytical guided wave propagation model and are very relevant to the authors’ work presented in the manuscript. The authors need to include those works in the introduction and point out the novelty and the difference between those work and the current manuscript.

  1. Equation (4) is not correct enough. It should be: , where is the distance from the excitation source to the damage, and  is the distance from damage to the receiver. Because there are two processes of the scatted signal: the first one is the process from the excitation source to the damage, and the second one is the process from the damage to the receiver, and the distance attenuation term should be the product of them. Although the process has a total propagation distance of , it is different from the one directly propagates a distance of .

  1. in the OMP algorithm, there is a parameter K, which determines the selected atoms from the dictionary. How does the authors select this parameter in application? It should be discussed.

There are quite a few minor errors in the English. Such as listed as follows:

-“different frequency” should be “different frequencies”, “different speed” should be “different speeds”;

-“the dispersed waves of each path are much easier to overlap with each other”, the authors use “easier” here, so compared with whom are they much easier to overlap?

-usually, “two modes of Lamb waves” are used instead of “two models of Lamb waves”;

-in Line 117, there should be no spaces before “where”.

-…..

Please check the manuscript once again.

Reviewer 2 Report

The present paper deals with an interesting method for defect localization throughout ultrasonic guided waves.

Besides the relevant content of the paper few issues must be adjusted.

>> Line 73: The authors mentioned Alguri et al. dictionary and mentioned Golato et al. dictionary. Later the author references another dictionary which is not clear in the text. The writing should be clearer about what dictionary the author is using or building.

>> Line 242 and 243 must be reviewed. Unexpected symbols.

>> All figure captions must be reviewed.

>> Fig. 14 is quite interesting, but all the red signs make it quite polluted. The image could be better explored in the text.

>> The results discussion must be improved.

Reviewer 3 Report

Article title: "Defect localization method based on spare sensor array and orthogonal matching pursuit decomposition"

The submitted paper deals with defect localization in shell structures using Lamb ultrasonic waves. This subject is of importance for structural health monitoring of airspace and naval transport and others. The authors provide a detailed and relevant bibliography. The theoretical background is explicitly presented and well supported by figure 2. The simulations and experiments are rigorously conducted and illustrated with artwork of quality. 

The paper is worth to be published in the MDPI Sensors journal; however, there are several minor remarks that authors are asked to address. 

The paper title seems to be too short of describing the work context. Proposal for the title: "Ultrasound defect localization in shell structures with Lamb waves using spare sensor array and orthogonal matching pursuit decomposition"

Figure 4. Provide distinguishable captions for (a) and (b) subfigures.

Figure 7: Correct the text: "Signal [be -> delete] separated by over-complete dictionnary" -> "Signal separated by over-complete dictionnary" 

Figure 11: "Experimental device" -> "Experimental setup"

Figure 13. Provide distinguishable captions for (d), (e), (i) and (j), subfigures.

Figure 14. Add captions for all (a) - (d) subfigures.

Conclusions. Please, include a statement, how the proposed OMP based algorithm can handle overlapping Lamb modes of order higher than A0 and S0.

Line 303: It seems that parentheses are missing for the xA and yA coordinates of point A. The same is for line 295

Line 351: "Take the collected signal" -> Rewrite all the phrases in the personal imperative form to phrases in passive voice or another impersonal form. This is appropriate to all text in the paper body but is not appropriate for bulleted algorithm descriptions as, for example, between lines 171 and 181. 

Typos and grammar issues:

43 In this paper, we try to decompose[d] the overlapped signals from the. The passive form of "decompose" is not necessary.

49 propagate with different speed[s], which cause[s] the wave packet [to] [elongated -> widen] during

97 composed and dispersion [is] removed by [the] OMP algorithm to [verifies -> verify] its correctness.

110 In other words, it prefers [remove "to"] selecting the set of atoms with the sparsest

143 The expressions of wave number, phase velocity[,] and group velocity are [respectively] [18]

179 Update the solution set { }, [Update -> and update] the residual...

Round 2

Reviewer 1 Report

The author has made major changes and responsed all my concerns.